# Health Behavior and Health and Psychosocial Planning for Retirement among Spanish Health Professionals

**DOI:** 10.3390/jcm7120495

**Published:** 2018-11-28

**Authors:** María Dolores Hurtado, Gabriela Topa

**Affiliations:** 1Hospital Virgen de las Nieves, Andalusian Health Service, 18014 Granada, Spain; mdhurtadomontiel@gmail.com; 2Department of Social and Organizational Psychology, National Distance Education University (UNED), 28040 Madrid, Spain

**Keywords:** psychosocial planning, health planning, retirement, health behavior

## Abstract

The aging of the workforce among health professionals demands attention to the study of their health behavior before retirement. The aim of the present study is to analyze the relationships between health planning and psychosocial planning—on the one hand—and health professionals’ health behavior, on the other. In addition, we will study the mediator role of public protection, self-insurance, and self-protection in the relationship between planning and health behavior. The sample includes 169 healthcare professionals from a public hospital in Spain. A serial mediation model estimating all of the parameters simultaneously was tested. The findings have confirmed the relationship between health planning and health behavior, as well as the serial mediation of the behaviors in this relationship. As the main causes of death in the Western world are not transmittable diseases, but cardiovascular diseases, diabetes, and other diseases that are closely linked to lifestyle factors, our findings strongly support that we are responsible for our long-term health status and well-being.

## 1. Introduction

During the past decades, the collective of health professionals in developed countries has aged, and this tendency is expected to continue [1,2,3,4]. Soon, the higher prevalence of occupational diseases and work-related health conditions among these workers may reduce their participation in employment, decrease their performance [5,6], and, at the same time, increase their personal health care expenditure and the public health expenditure [7]. Accordingly, the study of health behavior in the face of retirement in this collective is of special interest.

Health behavior includes all of the things that we do to positively influence our physical, mental, emotional, and psychological self [8]. As retirement approaches, health behavior seems especially relevant because this transition has implications in people’s general activity. Empirical evidence with general population [9] and some works with health professionals show the positive effects of planning the health behavior of people near retirement [10].

Planning consists of a lengthy process of identification of desires, needs, resources, and of the development and implementation of concrete plans [11]. Hence, planning can be focused on health itself—health planning—and on the activities and roles that the person will perform in the future —psychosocial planning. Moreover, such planning can be maintained by three behaviors [12] that consist of using public protection, developing concrete actions to consolidate self-insurance, and anticipating the emergence of problems by increasing one’s self-protection.

Diverse types of planning activities change the availability of resources in the physical, cognitive, financial, or social domains, which, in turn, seem to determine the quality of the person’s adjustment to the new situation [13]. In this sense, Yeung and Zhou [9] found that retirees with more planning activities receive more social support from their relatives and friends, which, in turn, promotes psychological well-being and satisfaction with life. This study aims to contribute to the literature in several ways. First, we analyzed and documented the relations between the antecedents of health behavior that have been little explored. Second, we focused on the mediator mechanisms through which planning influences behavior, that is, the possible forms of specific behaviors that people perform to protect themselves in the face of retirement.

Using this model, this study will analyze the relationships between health planning and psychosocial planning, on the one hand, and health behavior in Spanish health professionals over age 45, on the other. In addition, we will study the mediator role of public protection, self-insurance, and self-protection in the relationship between health planning and health behavior. A previous study analyzes these behaviors in a sample of nurses, but only to explore their antecedents, while considering public protection, private insurance, and self-protection as outcomes [10]. The results of this study will help to identify the most effective preparatory mechanisms and emphasize those areas that will have the greatest impact on health professionals’ long-term well-being. Individuals, organizations, and governments need substantiated information to design interventions that support health workers who are poorly prepared for retirement.

### 1.1. Health Behavior and Planning

Today, people are expected to understand that they are responsible for their health and to engage actively in its promotion. With this objective, a large amount of health information is available to the population in general and for health professionals [14,15,16]. The proximity of retirement offers a temporal indicator that can catalyze health behavior. People are being confronted with their own aging and they take note of how they can reduce their risks of developing serious illnesses and how to protect their health. Health behavior includes behaviors, such as regular exercise, healthy diet, and tobacco and alcohol reduction. Thus, health behavior becomes a desirable goal and an indicator of the overall well-being of the person at this vital stage [17].

Retirement planning seems to be synonymous with financial preparation, but this transition also implies significant changes at the level of health and daily activity that deserve to be addressed. Over the past ten years, there has been increasing awareness that planning occurs across many different domains beyond finances alone. More holistic models, such as that of Noone, Stephens, and Alpass [18], account for four key domains: financial, health, lifestyle, and the psychosocial planning process.

In short, retirement planning refers both to long-term efforts in preparation for health and to the creation of significant activities to be performed after retirement [19]. First, aging itself is associated with a set of age-related physical and cognitive changes, which are often accompanied by changes in emotion regulation [20], personality traits [21], among others. Hence, as poor health is a major determinant of early retirement [6], health planning has the capacity to prolong employment as well as to promote health behavior in later life. Second, since the loss of one’s job deprives one of a main activity in daily life, and this separates one from one’s economic rewards, recognition, social status, and relationships with other people, retirement planning should extend to the activities and social roles that one will play in the new situation. Psychosocial planning involves thinking about new roles, talking to those who have already retired about their experiences, and distancing oneself progressively from the role of worker [18]. However, it also implies managing new social relationships, finding activities—either remunerated or not—that occupy vital time and provide satisfaction. Psychosocial planning will allow one to maintain or increase the levels of mental, physical, and social development, providing a sense of competence and reinforcing one’s general functional capacity [22]. Based on the literature, in the present study proposes that: H1: Health planning will be positively related to health behavior. H2: Psychosocial planning will be positively related to health behavior.

### 1.2. Public Protection, Self-Insurance, and Self-Protection

In the 20th century, countries with a consolidated welfare state encouraged their citizens’ confidence in the social security systems as guarantees of the satisfaction of the needs of the population in old age. At the same time, it has been reported in the literature that people display unrealistic optimism and tend to overrate their future resources [23]. Hence, planning through public protection is characterized by the actions of obtaining information and resources for the future using exclusively the systems provided by the State, for example, managing a public health service, requesting Social security cards from the health center of the autonomous community, participating in programs of early detection of diseases, or participating in educational and recreational activities that are provided by the Public Health.

In the last decade, a debate has emerged about the limitations of this public system, and doubts about its capacity to meet the needs of a growing population of retirees. This has highlighted the need for people to increase their behaviors of self-insurance. Thus, the second mechanism, self-insurance, is retirement planning through concrete individual actions that contribute to generating resources and setting them aside to dispose of them to meet one’s personal needs in the future. Examples are taking out private life insurance or health insurance, investing in improving private health, regular medical check-ups, dental implants, or investing in training activities, leisure activities, or private social participation (formal education, language courses, private tourism).

Finally, the improvement of older people’s health behavior and the consequent reduction of health risks have revealed the importance of self-protection. Self-protection consists of preparing one’s future health through attention to nutrition, the adoption of healthy lifestyles, and formation in planning leisure time, involvement in voluntary work and activities to maintain one’s vital activity after ceasing to perform paid work. Epidemiological data have shown the relationship between regular exercise, healthy diet, and the reduction in mortality due to various causes [24].

According to empirical research [11], people can carry out various planning behaviors simultaneously or sequentially. In this sense, this research intends, firstly, to establish which planning behavior has the greatest impact on health behavior. Secondly, it intends to determine whether a certain sequential combination of these behaviors is more influential than another one.

In short, it is expected that: H3: The relationship between health planning and health behavior will be mediated sequentially by the behaviors of public protection, self-insurance, and self-protection. H4: The relationship between psychosocial planning and health behavior will be mediated sequentially by the behaviors of public protection, self-insurance, and self-protection.

## 2. Materials and Methods

### 2.1. Participants

The Bioethics Committee of the Junta de Andalucía (PEIBA) approved the project on October 27, 2015. The study sample includes 177 healthcare professionals of the public Hospital of Torrecárdenas (Almería, Spain). Mean years of professional tenure were 23.56 (standard deviation (SD) = 8.75). Sociodemographic characteristics of the sample are displayed in Table 1. Related to their professional categories, most of the sample were Nursing assistants (28.8%) and Registered Nurses (25.5%), Administrative staff were 10.7%, Occupational Therapists 0.6%; clinical Psychologist, 1,1%, Social workers 1,1%, and other categories (25.4%).

### 2.2. Instruments

Planning for retirement: We used two subscales of the Process of Retirement Planning Scales (PRePS), those related to Health planning (10 items) and Psychosocial planning (nine items) [18]. Both subscales showed appropriate psychometric properties in the study of adaptation to Spanish [25], with α = 0.77 and α = 0.76, respectively. Both the employed scales are rated on a five-point Likert-type response format, ranging from 1 (strongly disagree) to 5 (strongly agree). In the present study, reliability was α = 0.70 for both subscales. Examples of items were: “I think that it is still early to start thinking about my future health”; “I think it is worth finding new activities for retirement”; and, “I am prepared to reduce my working hours”.

Retirement planning scale (RPQII): The RPQII [12] presents 28 items, integrating three dimensions: Public Protection (five items, α= 0.85 in the original study), Self-insurance (13 items, α = 0.88, e.g.,), and Self-protection (nine items, α= 0.80). In the present study, we used the version that was adapted to Spanish by Topa, Segura, and Perez [10] in healthcare population. The scale is rated on a five-point Likert-type scale, ranging between 1 (very small amount of effort) to 5 (very large amount of effort). Reliability in this study was α = 0.89 (Public protection), α = 0.90 (Self-insurance), and α = 0.87 (Self-protection). Examples of items are: Public protection: “Participates in health programs subsidized by the State (e.g., health surveys, health assessments, model of pharmaceutical delivery, immunization programs)”; Self-insurance: “Takes out private health insurance”, “Participates in one or more workshops, seminars, and private courses on the planning of leisure, wellness, and health”; and, Self-protection: “Participates in one or more medical check-ups”.

Health behavior before retirement: this scale contains 15 items, focusing on changes in health behavior. It is rated on a five-point Likert type response format ranging from 1 (never) to 5 (always) [26]. Examples of items are: “Practices physical activity regularly”, “Makes appointments for medical examinations and check-ups”; “Takes development courses in another area with a view to a second professional stage”. In the present study, reliability was α = 0.87.

Sociodemographic data: We asked participants about their age, sex, education, organizational level in the present post, professional category, years of seniority in the post, work schedule, and type of contract.

### 2.3. Procedure

The data were collected between November and December 2015. A member of the research team went to the Hospital of Torrecárdenas to collect the data. The study was presented to potential participants, explaining its objective and guaranteeing anonymity in the treatment of the data. The participants signed an informed consent form to participate in the research and received a booklet that included the different scales of the study in a break of their work shift. The completed questionnaires were collected in a sealed envelope that was handed in to a member of the research team.

## 3. Results

### 3.1. Descriptive Analyses

Descriptive statistics for the study variables and correlation matrix are displayed in Table 2.

### 3.2. A Serial Mediation Model

A serial mediation model estimating all parameters simultaneously was tested while using the PROCESS macro for SPSS (SPSS version 25, SPSS Inc., Chicago, IL, USA). The hypothesized relationships were assessed using Model 6, which estimates the indirect effect of X (Health planning or Psychosocial planning) on Y (Health behavior) through M1 (Public protection), M2 (Self- insurance), and M3 (Self-protection). This procedure was based on 5000 bootstrap re-samples and it provided estimates of the indirect effect and associated confidence intervals. When zero is not included in the 95% bias-corrected confidence interval, it may be concluded that the parameter is significantly different from zero at *p* < 0.05. The direct effect of health planning on health behavior was significant but negative (c’ = −0.24, *p* = 0.04). The direct model was significant (F (4, 164) = 24.33, *p* < 0.001, *r*^2^ = 0.37). The overall effect was positive and statistically significant (c = 0.22, *p* = 0.01) and the overall model was significant (F (1, 167) = 7.99, *p* < 0.05, *r*^2^ = 0.05), only partially supporting Hypothesis 1 (Table 3).

The direct effect of psychosocial planning on health behavior was significant and positive (c’ = 0.36, *p* =0.00), as was the total effect (c = 0.48, *p* = 0.01), fully supporting Hypothesis (2 (Table 4). The direct model was significant (F (4, 164) = 28.23, *p* < 0.001, *r*^2^ = 0.41), as was the total direct model (F (1, 167) = 33.35, *p* < 0.001, *r*^2^ = 0.17).

To test Hypothesis 3, seven indirect effects were considered. Firstly, four of them were nonsignificant (Indirect effect 2, 3, 4, and 7), as shown in Table 3. Secondly, three of them were significant (indirect effect 1, 5, and 6). The indirect effects through Public protection (1), Self-insurance (5), and Self-insurance + Self-protection (6) displayed statistical significance, supporting Hypothesis 3. To test Hypothesis 4, seven indirect effects were considered (see Table 3). Five of them failed to reach statistical significance (Indirect effect 1, 2, 3, 4, and 7). Only the indirect effects of Psychosocial planning on health behavior through Self-insurance or both through Self-insurance and Self-protection were significant, thus partially supporting Hypothesis 4 (Figure 1). Although no specific hypotheses were made about the comparison between the various indirect effects, additional analyses were carried out. The mediation of Public protection in the relationship between health planning and health behavior remains significant, even when the indirect effects of Self-insurance and Self-protection are eliminated, separately or together (Table 4). In the relationship between psychosocial planning and health behavior, the comparison of the different effects showed that none of them were higher than the others.

## 4. Discussion

This research was designed with two main objectives: firstly, to analyze the relationship between health planning and psychosocial planning, on the one hand, and health behavior, on the other. Secondly, to analyze the serial mediation of the behaviors of public protection, self-insurance, and self-protection in these relationships. The findings have confirmed the relationship between health planning and health behavior, as well as the serial mediation of the behaviors in this relationship. Especially outstanding is the influence of the behaviors that are based on public protection, which confirms earlier findings about people’s optimism about their retirement and their confidence in public protection as the guarantor of their future well-being [10]. Psychosocial planning, on the other hand, influences health behaviors mainly through self-insurance and self-protection activities. Although some recent reviews [27] seem to support the idea that preparatory activities are not among the best predictors of adjustment to retirement, most of the empirical evidence shows the positive impact of planning on subsequent well-being. The recent study of health behavior has stressed the impact of different predictors, such as anticipated regret, a diagnosis of illness, or very stressful life events [28], like the death of a spouse [29], and the paradoxical role of other antecedents, such as perceived control [30]. In addition, numerous meta-analyses of the empirical research have been conducted to date [31,32]. However, there are few works analyzing health behaviors in the face of retirement, even though this event represents a significant change in all life areas. While health behavior research has focused on anticipated regret as a novel risk appraisal [33], the perception of retirement as an abrupt change that puts an individual’s health at risk has received less attention. 

In this sense, it seems especially relevant to consider the influence on health behavior, not only of actual health planning itself, but also of planning the use of time and daily activity patterns, while considering their relationship with work and retirement. In this sense, some studies [34] pointed out that the ability to control one’s time has proven to be one of the best predictors of the quality of the transition to retirement. The use of time is an essential component of the experience of retirement that people must face. In this sense, the influence of the individual experiences of aging should be included among the emotional antecedents of health behavior in older adults [35]. The fact that the studies on health planning in the face of retirement in healthcare population are still scarce deserves special consideration. However, health planning and psychological planning are not only important for the personal well-being of health professionals, but also for their impact on patient outcomes. Research has confirmed that patient outcomes are influenced or impacted by the welfare of the doctor [36]. The evidence suggests that doctors’ adjustment to subsequent professional transitions can facilitate retirement planning [37].

Our findings have specific implications both for late career management plans and for the design of supportive policies for older workers. At the individual level, under a preventive approach, health professionals should invest time and resources on decision making about the final steps of their careers [38,39,40]. As their implicit knowledge and expertise have been accumulated during their professional development, management plans can include non-profit activities helping others, continuing education, or entrepreneurship initiatives. At the organizational level, programs for enhancing health literacy would be useful for workers approaching retirement. In the same vein, due to the strong connection between health and wealth, financial planning for retirement should be considered in its relationship with health planning for retirement [41].

The present study has various limitations that should be mentioned. First, the small sample size and its lack of representativeness in the population of health professionals. Despite that the data was collected in a single health center to homogenize the sample, the findings cannot be extrapolated to the rest of the health professionals. Second, the study variables were measured through self-reported questionnaires, and the data were collected at a single time. Health behavior was evaluated in general, rather than attending to specific behaviors within a domain, such as physical activity and diet. Domain-specific results could vary significantly, as suggested by [42]. However, recent works have explored the common antecedents for health behaviors, such as motivation, self-efficacy beliefs, attitudes, and social influences with important predictive power over a wide range of health behaviors [43].

Third, the potential influence of extraneous variables that were not assessed, such as personal, social, and economic circumstances, etc., which can directly influence or alter the meaning each person assigns to retirement should be considered when interpreting these findings [36]. To develop effective interventions to facilitate health behavior, a detailed understanding of the demographic, psychological, and ecological factors is required, as suggested Hardcastle and Hagger [44]. In this sense, technologies offer today new ways for implementing non-face to face care management for elderly population [45]. Finally, we should not ignore the fact that the psychosocial environment also includes the family and the partner, who also seems to be an important predictor to explain health behavior in mature ages [46]. As recent studies suggested, from the Family-centered perspective, interventions that are aimed to increase self-efficacy and to empower families’ support can be successful for people with chronic conditions [47].

## 5. Conclusions

Planning is relevant to the training of future health professionals, who could develop early on the skills needed to become referents of health and well-being and can give appropriate advice to their patients in these matters. Among the strategies that are available to plan leaving the working world, we suggest flexible working hours, the reduction of barriers to reconcile work and family, and the increase of facilities to prioritize health [46], such as bridge employment opportunities and programs to help doctors to change the content of their work [47]. A change of tasks to nonclinical dimensions, such as teaching or tutoring, could help to retain older health professionals, while facilitating the transfer of knowledge to younger professionals. In the same vein, training in selection, optimization, and compensation skills could help older workers to make plans better adapted to their wishes and needs, clarifying their objectives and developing optimal means to achieve the goals that they have set [20]. Finally, to synthetize the meaning and use of this research we could suggest at least to ways. First, our findings highlighted the relevance of health planning and psychosocial planning to change health behaviors of older workers, and, in turn, positively influence their future wellbeing. Second, this empirical evidence tries to underline the urgency of health behavior changes in order to reduce future health costs for aged societies and better health protection for individuals.

## Figures and Tables

**Figure 1 jcm-07-00495-f001:**
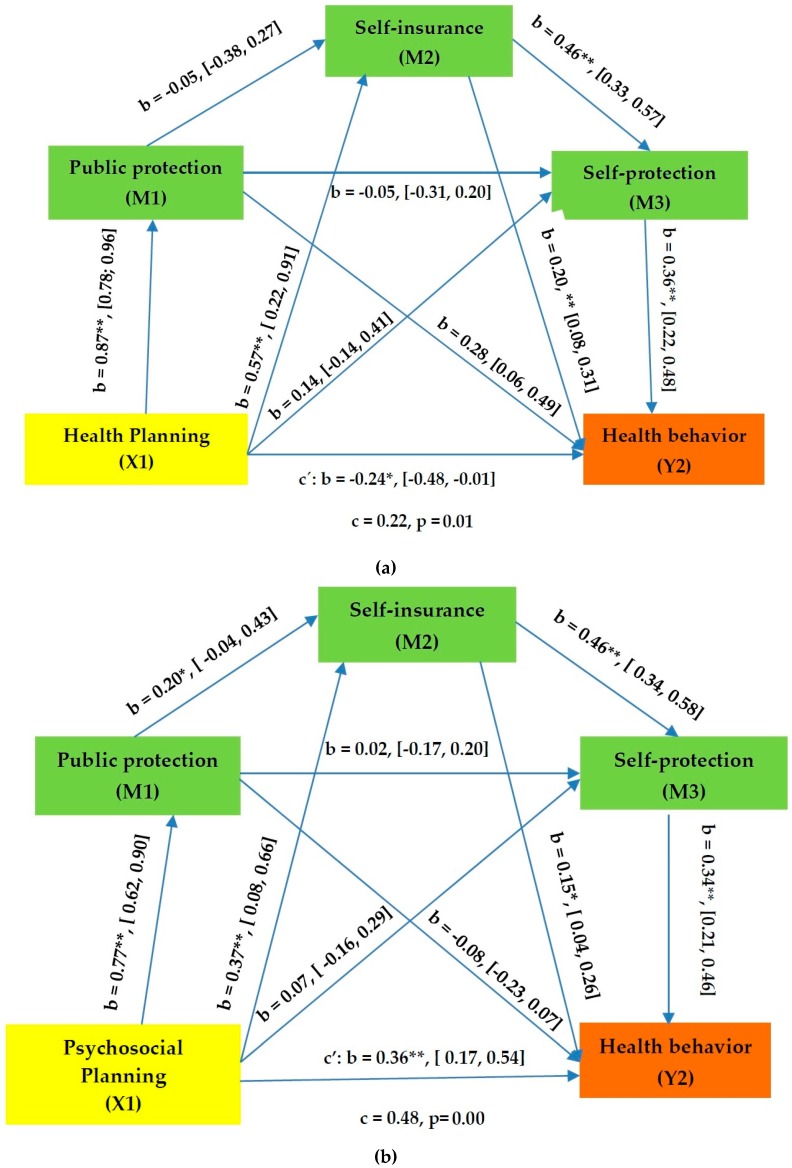
(**a**) Serial mediating effects of Public Protection, Self-insurance and Self-protection in the relationships between Health Planning and Health Behavior (**b**) Serial mediating effects of Public Protection, Self-insurance and Self-protection in the relationships between Psychosocial planning and Health Behavior. Note: B: unstandardized coefficients and 95% Confidence Interval; **p* < 0.05; ***p* < 0.01.

**Table 1 jcm-07-00495-t001:** Sociodemographic Characteristics.

Characteristics (Percentages)	
Gender	
Men	27.1%
Women	71.2%
Age	
>50	48.6%
<50	47.5%
Educational level	
Basic studies	9.6%
Professional Training	29.9%
Bachelor’s degree	13.6%
University Degree	41.2%
Others	4.5%
Type of shift	
Full time	92.7%
Part time	5.1%
Others	0.6%
Type of employment contract	
Fixed	45.8%
Temporary	9.0%
Civil servants	27.7%
Eventual	17.5%

**Table 2 jcm-07-00495-t002:** Descriptive statistics and correlation matrix.

Variables	M	SD	1	2	3	4	5	6
1. Health Planning	3.37	0.64		0.61 **	0.83 **	0.27 **	0.28 **	0.19 **
2. Psychosocial planning	3.20	0.56			0.64 **	0.24 **	0.34 **	0.39 **
3. Public protection	3.74	0.67				0.21 **	0.30 **	0.23 **
4. Self-protection	3.73	0.77					0.55 **	0.55 **
5. Self-insurance	2.84	0.87						0.48 **
6. Health behavior	2.89	0.68						

*n* = 169, M = mean, SD = standard deviation, ***p* < 0.01.

**Table 3 jcm-07-00495-t003:** Direct, total and indirect effects.

**Health Planning and Health Behavior**	**b**	**Boot SE**	***t***	**95% CI**
Direct Effect Health Planning → health behavior	−0.24 **	0.12	−2.0	(−0.48, −0.008)
Total Effect Health Planning → health behavior	0.22 **	0.08	2.82	(0.07, 0.37)
IE 1: Health Planning →Public protection→ health behavior	0.24	0.09		(0.07, 0.41)
IE 2: Health Planning → Public protection→ Self-insurance → health behavior	−0.01	0.03		(−0.07, 0.05)
IE 3: Health Planning → Public protection→ Self-protection → health behavior	−0.02	0.05		(−0.11, 0.07)
IE 4: Health Planning →protection→ Self- insurance →Self-protection → health behavior	−0.01	0.02		(−0.06, 0.04)
IE 5: Health Planning → Self- insurance → health behavior	0.11	0.05		(0.04, 0.24)
IE 6: Health Planning → Self- insurance → Self-protection → health behavior	0.09	0.04		(0.04, 0.18)
IE 7: Health Planning → Self-protection → health behavior	0.05	0.05		(−0.04, 0.15)
**Psychosocial Planning and Health Behavior**	**b**	**Boot SE**	***t***	**95% CI**
Direct Effect: Psychosocial Planning → health behavior	0.36 **	0.09	3.77	(0.17, 0.54)
Total Effect: Psychosocial Planning → health behavior	0.48 **	0.08	5.77	(0.32, 0.65)
IE 1: Psychosocial Planning → Public protection→ health behavior	−0.06	0.06		(−0.18, 0.05)
IE 2: Psychosocial Planning → Public protection → Self-insurance → health behavior	0.02	0.02		(−0.01, 0.07)
IE 3: Psychosocial Planning → Public protection → Self -protection → health behavior	0.00	0.03		(−0.04, 0.06)
IE 4: Psychosocial Planning → Public protection → Self-insurance → self. protection→ health behavior	0.02	0.02		(−0.00,0.07)
IE 5: Psychosocial Planning → Self-insurance → health behavior	0.06 **	0.03		(0.01, 0.14)
IE 6 Psychosocial Planning → Self-insurance → Self -protection → health behavior	0.06 **	0.03		(0.01, 0.13)
IE 7: Psychosocial Planning → Self -protection → health behavior	0.02	0.04		(−0.06, 0.10)

Note: *n* = 169; IE: indirect effect, SE = Standard Error, CI = Confidence interval. Sample size Bootstrap for Indirect Effects = 5000; b = non-standardized regression coefficient; ***p* < 0.01.

**Table 4 jcm-07-00495-t004:** Contrasting indirect effects between health planning and health behavior.

	b	SE	95% CI
C1: Indirect 1 minus indirect 2	0.25	0.09	(0.07, 0.43)
C2: Indirect 1 minus indirect 3	0.25	0.10	(0.05, 0.44)
C3: Indirect 1 minus indirect 4	0.25	0.09	(0.07, 0.42)

Note: *n* = 169, C1: indirect contrast; CI = Confidence interval; SE = Standard Error; b = non-standardized regression coefficient. Indirect 1: Health Planning → Public protection → health behavior; Indirect 2: Health Planning → Public protection → Self- insurance → health behavior; Indirect 3: Health Planning → Public protection → Self- protection → health behavior; Indirect 4: Health Planning → Public protection → Self- insurance → Self-protection → health behavior.

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
