# Peer review of "Health Behavior and Health and Psychosocial Planning for Retirement among Spanish Health Professionals"

_jcm, 2018, doi:10.3390/jcm7120495_

Reviewer 1 Report

The content and title of the paper do not match. It analyzes the relationship between health behavior and health planning and psychosocial planning, not occupational disease.

It is necessary to interpret the meaning of the result in various ways, for example, management plans, policy support directions, and so on. It is difficult to understand the meaning and use of this paper.

Author Response

Dear reviewer, 

following your recommendations and the Editor suggestions, we have changed the manuscript and prepared a point by point answer. (please, see the attached file).

Thank you, 

the authors

Reviewer 2 Report

It was a pleasure to review the above manuscript pertaining to health behaviors of health professionals facing retirement. The investigation is very interesting and relevant to many.  The study was well designed, executed, written-up and presented for publication. I recommend that the manuscript be published with minor revisions. Please see my comments and recommendations below.

First impression:

The manuscript is well written, interesting and a valuable contribution to the social science literature pertaining to and behavioral responses to major life events such as retirement. The implications are significant to the health industry. The manuscript contains information that will be appreciated and useful to health care professionals, health care economists, health care scientists, and workers.

Strengths:

Well designed cross-sectional investigation. Rigorous methods, appropriate data collection, analysis and conclusions were executed. Well written, organized and presented manuscript. Tables and figures were appropriate, and value added.

Weakness:

The only weakness that I noticed was missing labeling on Table 1 for education. The percentiles are offered but no corresponding classification of years of education. I think that this may be a print error.

Author Response

Dear reviewer, 

following your recommendations and the Editor suggestions, we have changed the manuscript and prepared a point by point answer. (please, see the attached file).

Thank you, 

the authors

Round  2

Reviewer 1 Report

I want you to exclude occupational diseases and prevention of work-related risk factors from the keywords.

Specific phrases such as on the one hand are repeated in the text. I hope these phrases are properly modified.

Author Response

Dear Reviewer, thank you very much for your careful reading of our paper. 

We have tried to avoid repetition of phrases. 

We have uploaded a new version, also excluding the keywords that you suggested. 

Thank you for your effort.